# Effect of Preharvest Application of CPPU and Perforated Packaging on the Postharvest Quality of Red-Fleshed Pitaya (*Hylocereus polyrhizus* sp.) Fruit

**Pai-Tsang Chang**

Department of Horticultural Science, National Chiayi University, 300 Xuefu Rd., Chiayi City 60004, Taiwan; ptchang@mail.ncyu.edu.tw

**Abstract:** The objective of this study was to investigate the effect of the preharvest application of forchlorfenuron (CPPU) and perforated polyethylene bag packaging (PPE) on maintaining the postharvest quality of red-fleshed cv. 'Da-Hong' pitaya (*Hylocereus polyrhizus* sp.) fruit. On the flowering day, 100 mg·L$^{-1}$ CPPU was sprayed on the bracts and water was used as the control. After harvest, all fruits were divided into three package treatments, which were packed without bags, packed with and without PPE bags, and stored at 5 ± 0.5 °C and 90 ± 5% relative humidity for 21 days, followed by 7 days at 20 °C and 75 ± 5% relative humidity without bags for quality evaluation. Significantly higher bract thickness (2.26 vs. 1.44 mm), longer fruit length (120.5 vs. 109.04 mm), and greater firmness (1.56 vs. 1.04 kg·cm$^{-2}$) were recorded for the CPPU treated fruit at harvest. Preharvest application of CPPU with perforated packaging resulted in significantly greener bracts, a lower yellow index, fewer chilling incidences, and a lower decay ratio, but there was a slight decrease in respiration rate during cold storage at 5 °C for 21 days. However, all criteria reached the threshold when fruits were transferred to 20 °C for 7 days. In conclusion, preharvest CPPU application plus perforated packaging is the best combination for the long-term storage of red-fleshed pitaya fruit at 5 °C.

**Keywords:** plant growth regulator; dragon fruit; plastic bag; polyethylene bag

## 1. Introduction

Red-fleshed pitaya (*Hylocereus polyrhizus* sp.) is a non-climacteric fruit newly grown in many countries because of its attractive color and nutritional properties [1–3]. The flesh is red-purple in color a and has sweet taste, and it is consumed as fresh fruit or processed as juice, jellies, and beverages [4]. When horticultural crops are harvested, the first postharvest challenge is water loss through transpiration, resulting in reduction in quality and storage period [5]. Once harvested, pitaya fruit that are not properly handled and stored experience postharvest problems such as bract withering, bract browning, and water loss, leading to distraction and affected marketability [6].

Alternatively, a lower storage temperature is applied effectively for reducing respiration rates, water loss, ethylene sensitivity, and decay occurrence, resulting in a longer storage life for fresh horticultural products [7]. However, pitaya fruits are very susceptible to chilling injury (CI) if they are stored at 0 °C for 12 to 15 days, or for up to 21 days at 6 °C, leading to fruit decay, weight loss, bract browning, peel desiccation, and fruit softening [6,8,9]. Thus, a storage temperature between 5 °C and 10 °C is recommended to avoid chilling injury depending on pitaya fruit varieties [8,10–13].

Previous studies have been carried out to enhance the storage life and maintain the quality of pitaya fruit in relation to low temperature storage using different strategies such as chitosan coating, postharvest hot air treatment, packaging, and preharvest spraying with forchlorfenuron (CPPU) [10–12,14,15]. Among the above strategies, polymeric bags were thought to create a low oxygen ($O_2$) and high carbon dioxide ($CO_2$) concentration, causing

a reduction in the respiration rate and water transpiration, and so have been widely used in packaging for fresh fruit and vegetables in postharvest storage [16–18]. However, several studies have also indicated that polymeric bags represent a limitation in permeability level of $O_2$ compared to $CO_2$, which shorten goods' storage life and quality [19–21].

Thus, perforated plastic bags are a proposed technology in postharvest packaging because of their higher exchange rates of gases (particularly $O_2$ and $CO_2$) and water vapor. Previous studies have shown that perforation-mediated modified atmosphere packaging (PM-MAP) has been applied in postharvest technology for the retention of fruit quality at low storage temperatures [16,22–24]. However, limited studies have reported that perforation packaging is beneficial for the reduction of water loss and peel shriveling in yellow pitaya (*Selinicereus megalanthus* Shuman) [15] and in white-fleshed pitaya (*Hylocereus undatus*) [10]. In addition, the forchlorfenuron (*N*-[2-chloro-4-pyridyl]-*N'*-phenylurea) (CPPU), which was first applied on red-fleshed pitaya flowers to maintain the appearance and to delay chilling injury incidence when fruits were stored at 5 °C [12], mimics cytokinins (CKs) activity. The objective of this study was therefore to determine the effect of the preharvest application of CPPU in combination with the perforated polyethylene (PPE) bag to extend the storage and shelf-life quality of red-fleshed pitaya cv. 'Da-Hong'.

## 2. Materials and Methods

### 2.1. Field Experiment

Red-fleshed pitaya (*Hylocereus polyrhizus* sp.) cv. 'Da-Hong' grown at commercial orchard located in Tungshan township, Tainan city, Taiwan (lat. 23.18° N, long. 120.25° E, elevation 50 m) was selected in the 2019 season. Each plant within 2 year old cladodes was randomly marked and managed to produce one flower only for the experiment. Aqueous solution was prepared by diluting CPPU solution (Sumitomo Chemical, Tokyo, Japan) at a concentration of 100.0 mg·$L^{-1}$; water was used as the control. On the flowering day, 4.0 mL of CPPU and water were sprayed, respectively, one time per flower using a nozzle sprayer.

Red-fleshed pitaya fruits (*Hylocereus polyrhizus* sp.) were hand-harvested 32 days after anthesis (DAA) and transported to the laboratory immediately. The fruits were visually assessed as being without damage or disease, and then tap-water washed and air-dried at 25 °C for 24 h before packaging.

### 2.2. Perforation Packaging and Storage

Red-fleshed pitaya fruits were randomly divided into six lots as control without packaging, control without perforated packaging, control with perforated packaging, CPPU without packaging, CPPU without perforated packaging, and CPPU with perforated packaging. The bag was low density polyethylene single layer of 35 cm in length, 25 cm in width, and 0.03 mm in thickness. The perforated holes were 1 cm in diameter with 45 holes·$m^{-2}$. Fruits were stored at 5 ± 0.5 °C and 90 ± 5% relative humidity for 21 days, and then removed to storage at 20 °C and 75 ± 5% relative humidity for seven days without packaging. Each lot was comprised of four boxes (replications) with ten fruits per replication.

### 2.3. Fruit Weight Loss Percentage

Fruit weight was recorded using a digital balance (SCALTEC SBA-51, Goettingen, Germany) every 7 days during the 21-day storage; thereafter, during the shelf-life the fruit weight was recorded every day for another 7 days. The weight loss percentage was calculated with the following Equation (1):

$$\%WL = ((Wa - Wf)/Wa) \times 100\% \tag{1}$$

where %WL = weight loss percentage, Wa = fruit weight after harvest, Wf = final fruit weight at the indicated period.

### 2.4. Fruit Size

The fruit size was determined by measuring the longitudinal length (L) (cm) and transverse length (T) (cm) with a digital electronic calipers ruler (Mitutoyo 530–115, Kawasaki, Japan). Four fruits per box were randomly picked from each treatment, and the results were expressed as an average of four replicates.

### 2.5. Fruit Firmness

The above four fruits were used for firmness. Each fruit was placed on a stand, and a penetrating probe (6.0 mm) at $0.5 \ mm \cdot cm^{-1}$ speed coupled with a texture analyzer (Stevens-LFRA, Middleboro, MA, USA) was used to puncture 2.0 mm depth on each opposite side at a point around the equator. The readings were averaged in units as $kg \cdot cm^{-2}$.

### 2.6. Bract Thickness

The fruit used for firmness were also measured its bract thickness by using a digital electronic calipers ruler (Mitutoyo 530–115, Kawasaki, Japan). The third bracts from the end of the fruit stem were taken from the sunny side and back side. The average readings were expressed as the thickness of four replications.

### 2.7. Bract Browning

Bract browning (BB) was visually assessed for severity. The BB evaluation was based on five levels, ranging from 1 to 5 as follows: 1~1.9 = none; 2~2.9 = slight; 3~3.9 = moderate; 4~4.9 = moderately severe; $\geq$5 = severe [12]. The BB assessments were recorded after 7, 14, and 21 days of storage, and every day during the shelf-life on sixteen fruits from each treatment. The results were expressed as a yellow index calculated using the following Formula (2):

$$\Sigma((n \times B_1) + \ldots + (n \times B_5))/(N) \times 100\% \tag{2}$$

where n is the number of browning fruits per level; Bx is the level of bract browning; and N is the total number of fruits examined.

### 2.8. Chilling Injury

During storage, each fruit was visually assessed for the presence of chilling injury (CI) when the pericarp exhibited brown spot. The severity of pericarp browning was calculated as 0 = no brown spot; 1 = 1–2 brown spots; 2 = some brown spots, limited marketability; 3 = 50%; 4 = 75% or entire fruit browning. The CI index was calculated using the following Formula (3):

$$(browning \ scale \times proportion \ of \ corresponding \ fruit) \ within \ each \ class \tag{3}$$

### 2.9. Decay Index

During the storage and shelf-life, decay severity was visualized depending on decay region: 0 = 0%; 1 = 1–5%; 2 = 6–10%; 3 = 11–20%; 4 = 21–30%, 5 more than 30%. The decay severity was expressed as follows (4):

$$\Sigma((n \times D_0) + \ldots + (n \times D_5))/(N \times D_5) \tag{4}$$

where n is the number of decayed fruits per level; Dx is the grade of decay; and N is the total number of fruits examined multiplied by the maximum numerical decay grade, i.e., 5.

### 2.10. Respiration Rate

During storage, one fruit without visible symptoms was taken from each replication and put into a sealed respiration cylinder (2.0 L) for 2 h for respiration measurement. The gas was collected using a 1.0 mL syringe and injected into a GC-8A gas chromatograph (Shimadzu, Kyoto, Japan) for $CO_2$ content. The respiration rate was calculated based on the amount of $CO_2$ produced per fruit fresh weight per hour.

*2.11. Total Soluble Solid, Titratable Acid, and TSS to TA Ratio*

The same four fruits used for firmness measurement were cut into 2.0 cm thick wedges longitudinally along the central axis. The top, central, and bottom flesh was punched using a diameter 1.5 cm straw, and three samples were placed on two layers of 5.0 × 5.0 cm gauze and squeezed; a few drops of juice from each fresh pitaya fruit were placed on a digital refractometer (ATAGO PAL-1, Tokyo, Japan) for total soluble solid (TSS) expressed as a °Brix value. Other juice samples were titrated with 0.1 N NaOH to pH 8.2 as titratable acid (TA) [25]. The TSS divided by the corresponding TA value was represented as TSS/TA.

*2.12. Statistical Analysis*

The analysis of variance (ANOVA) was carried out by SAS (version 9.2; SAS Institute, Cary, NC, USA). The mean values for four replications were performed using a least significant difference (LSD) test at a 5% significance level ($p \leq 0.05$).

## 3. Results

*3.1. Fruit Physicochemical Characteristics*

Plants treated with 100.0 mg·L$^{-1}$ CPPU resulted in significantly longer fruit length (120.5 mm), more thickness (2.26 mm), and firmness (1.56 kg·cm$^{-2}$) than the control (109.4 mm in length, 1.44 mm in thickness, and 1.04 kg·cm$^{-2}$ in firmness) (Table 1). However, the CPPU did not show a significant effect on fruit width (W) or fresh weight (FW) (Table 1). Although CPPU had no effect on fruit TSS and TA when compared to the control fruit, the treated fruit had a higher TSS/TA ratio (Table 1).

**Table 1.** The effect of CPPU on physico-chemical characteristics of red-fleshed pitaya at harvest.

| Treatment | Physico-Chemical Characteristics | | | | | | | |
|---|---|---|---|---|---|---|---|---|
| | Weight (g) [z] | Length (mm) | Width (mm) | Bract Thickness (mm) | Firmness (kg·cm$^{-2}$) | TSS (°Brix) | TA (%) | TSS/TA |
| Control | 664.6 ± 23.5a [y] | 109.4 ± 3.3b | 99.6 ± 1.5a | 1.44 ± 0.1b | 1.04 ± 0.0b | 14.9 ±0.3a | 0.24 ± 0.0a | 62.1 ± 2.4a |
| CPPU | 671.7 ± 30.2a | 120.5 ± 2.4a | 102.0 ± 2.1a | 2.26 ± 0.1a | 1.56 ± 0.0a | 15.3 ±0.3a | 0.23 ± 0.0a | 66.5 ± 3.6a |

[z] Each value represents means of four replications ± SD. [y] Means with the same letter within column are not significantly different by LSD test at 5%.

During 5 °C storage, CPPU treated fruits could have significantly higher thickness and firmness than the control fruit either with or without PE bag packaging, but there were no significant differences among other criteria. Although all physico-chemical characteristics decreased when all fruit were moved to 20 °C storage, the bract thickness and firmness were higher in the CPPU treated fruit than in the control fruit regardless of packaging (Table 2).

*3.2. Weight Loss*

Fruit weight loss increased during storage; however, fruit without packaging had significantly more weight loss from 1.0% to 3.5% either with or without CPPU treatment (Figure 1). In all PE bag treatments, both the control and CPPU treated fruit packed without perforated plastic bags had the lowest weight loss between 0.1 and 0.3% during 5 °C storage. On the other hand, fruit packed with perforated plastic bags had slight weight loss between 0.3 and 0.4% (Figure 1). When fruits were transferred to 20 °C storage, the weight loss increased dramatically, and the highest weight loss was observed in fruit that were treated with CPPU preharvest and stored without packaging (6.5%). Although weight loss was found in all treatments, fruits stored at 20 °C resulted in more weight loss than those stored at 5 °C (Figure 1).

**Table 2.** The preharvest application of CPPU and perforation packaging on quality of red-fleshed pitaya during storage and shelf-life.

| Treatment | | | Physico-Chemical Characteristics | | | | | | | | | |
|---|---|---|---|---|---|---|---|---|---|---|---|---|
| | | | 21 Days 5 °C | | | | | 7 Days 20 °C | | | | |
| | | | Bract [z] Thickness (mm) | Firmness (kg·cm$^{-2}$) | TSS (°Brix) | TA (%) | TSS/TA | Bract Thickness (mm) | Firmness (kg·cm$^{-2}$) | TSS (°Brix) | TA (%) | TSS/TA |
| Control | N [x] | | 1.24 ± 0.1b [y] | 1.00 ± 0.0b | 13.2 ± 0.2a | 0.2 ± 0.0a | 66.1 ± 2.6a | 0.8 ± 0.1b | 0.5 ± 0.0b | 13.5 ± 0.4a | 0.2 ± 0.0a | 67.1 ± 2.5a |
| | 0 | | 1.34 ± 0.1b | 1.02 ± 0.0b | 13.6 ± 0.3a | 0.2 ± 0.0a | 68.0 ± 1.4a | 0.83 ± 0.1b | 0.4 ± 0.0b | 13.5 ± 0.3a | 0.2 ± 0.0a | 66.8 ± 3.6a |
| | 2 | | 1.34 ± 0.1b | 1.01 ± 0.0b | 13.4 ± 0.7a | 0.2 ± 0.0a | 67.4 ± 1.1a | 0.9 ± 0.1b | 0.4 ± 0.0b | 13.5 ± 0.4a | 0.2 ± 0.0a | 67.3 ± 4.6a |
| CPPU | N | | 1.91 ± 0.1a | 1.22 ± 0.0a | 13.5 ± 0.7a | 0.2 ± 0.0a | 67.5 ± 3.1a | 1.5 ± 0.3a | 1.01 ± 0.0a | 13.5 ± 0.4a | 0.2 ± 0.0a | 67.5 ± 3.4a |
| | 0 | | 2.20 ± 0.2a | 1.28 ± 0.0a | 14.5 ± 0.2a | 0.2 ± 0.0a | 71.5 ± 3.1a | 1.9 ± 0.3a | 1.03 ± 0.0a | 14.1 ± 0.2a | 0.2 ± 0.0a | 68.5 ± 4.4a |
| | 2 | | 2.01 ± 0.3a | 1.21 ± 0.0a | 14.0 ± 0.5a | 0.2 ± 0.0a | 70.2 ± 2.2a | 1.8 ± 0.2a | 1.02 ± 0.0a | 14.0 ± 0.4a | 0.2 ± 0.0a | 68.8 ± 4.8a |

[z] Each value represents means of four replications ± SD. [y] Means with the same letter within column are not significantly different by LSD test at 5%. N [x]: Fruit without packaging; 0: Fruit packed without perforation package; 2: Fruit packed with perforation package.

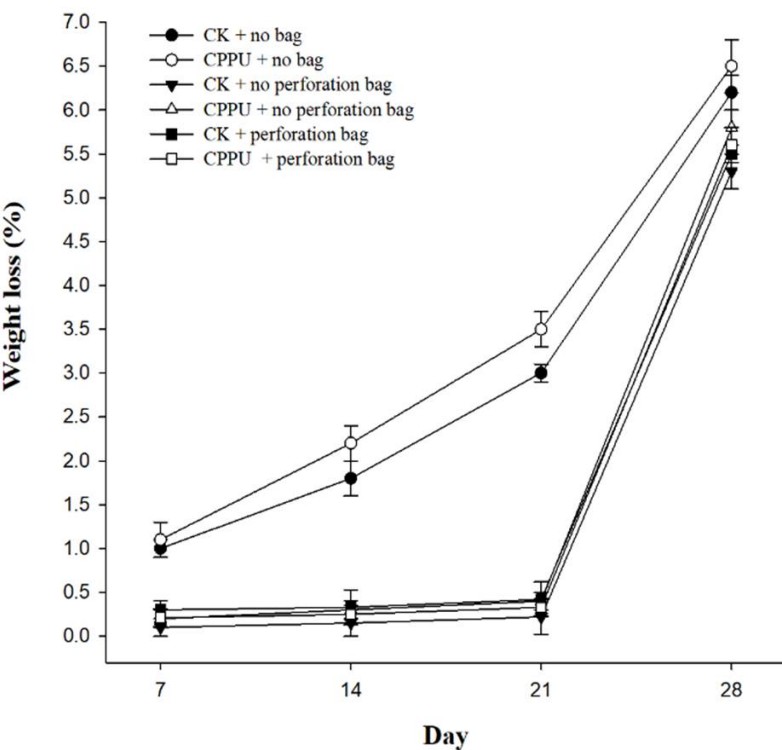

**Figure 1.** The effect of preharvest application of CPPU and perforation packaging on weight loss (%) at 5 °C for 21 days and then removed to 20 °C for 7 days. Bars represent means of four replications ± SD (*n* = 4).

### 3.3. Bract Browning

The fruit treated with CPPU showed significantly more vivid-green bracts compared with the control fruit (Figure 2A), and had a 1.1 yellow index compared to the control fruit (1.55) at harvest (Figure 2B). The control without packaging showed the highest yellow index at 5.0, and the CPPU treated fruit with perforated packaging exhibited the lowest yellow index on the 21st day. On the other hand, the control fruit showed withered bracts with a 5.5 yellow index, and those treated with CPPU could maintain slightly greener bracts of between 4.0–4.2 yellow index during the 20 °C storage period.

### 3.4. Chilling Injury and Decay Severity

A minor chilling injury (CI) symptom occurred in both the control and the CPPU treated fruit at 5 °C for 21 days (Figure 3A), and severer CI was exhibited in all fruit when moved to 20 °C for 7 days (Figure 3B). Besides, decay symptoms occurred accompanied with the storage experiment; fruit with or without preharvest CPPU treatment showed the lowest decay index (DI) (0.6–0.8) without packaging at 5 °C on the 21st day, and had 2.8 and 2.2 in the control and the CPPU treated fruit at 20 °C for 7 days, respectively. On the other hand, fruit packed without perforated PE bags resulted in 2.0–2.8 DI during 21 days storage at 5 °C while the control and the CPPU treated fruit reached severe DI of 6.8 and 5.8 during 20 °C storage period, respectively. When fruits were packed with perforated PE bags the DI was expressed between 1.0–1.2 after 21 days of storage at 5 °C. However, the control fruit had 3.7 DI and the CPPU treated fruit showed 3.0 DI when they were removed to 20 °C for 7 days (Figure 4).

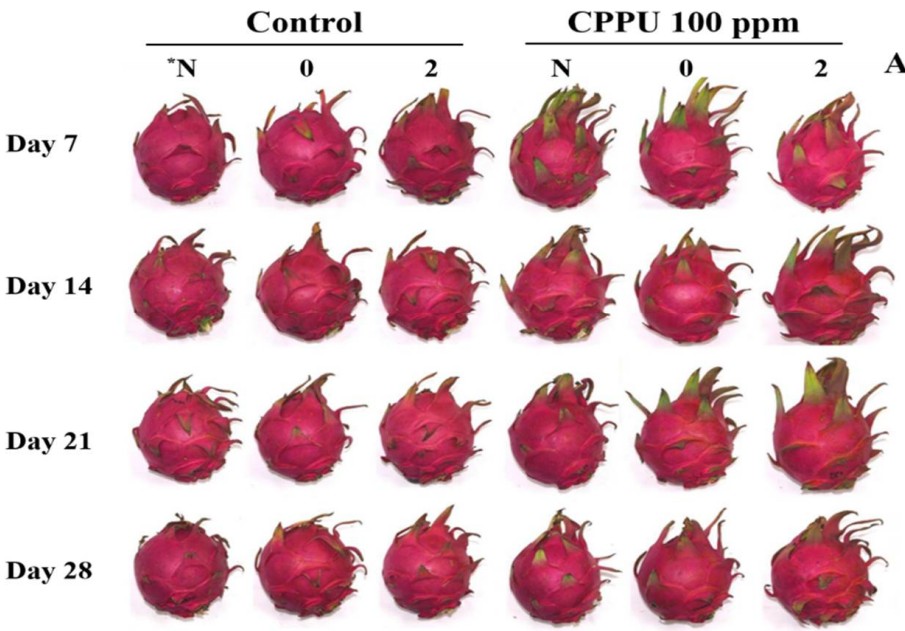

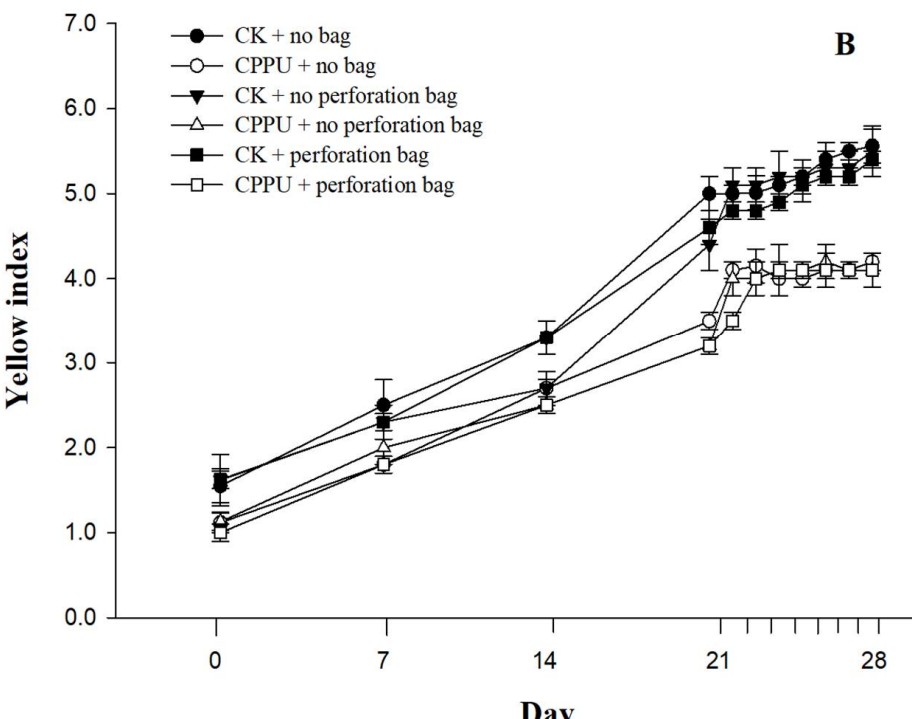

**Figure 2.** The effect of preharvest application of CPPU and perforation packaging on bract browning of 'Da Hong' (**A**) and yellow index (**B**) during postharvest storage at 5 °C, and transferred to 20 °C for 7 days. Bars represent means of four replications ± SD (*n* = 4). *N: Fruit without packaging; 0: Fruit packed without perforation; 2: Fruit packed with perforation.

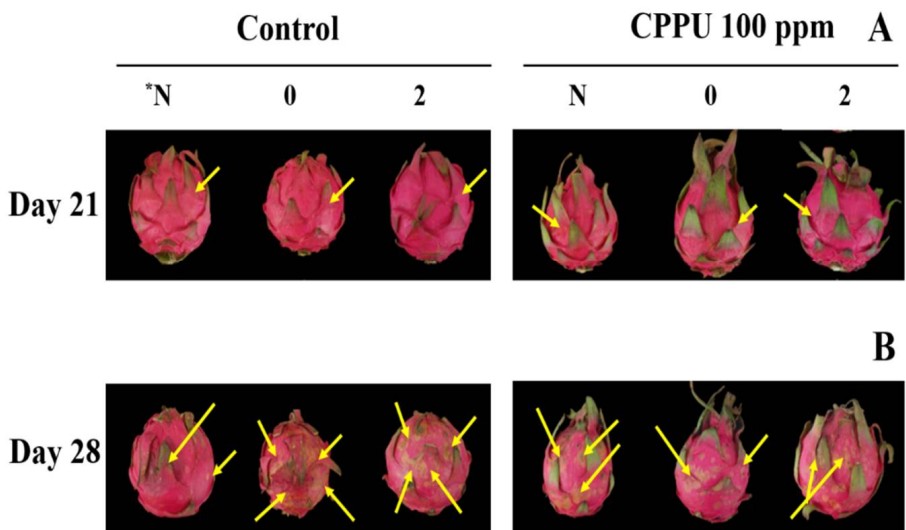

**Figure 3.** The effect of preharvest application of CPPU and perforation packaging on chilling injury (CI) incidence of red-fleshed pitaya 'Da Hong' fruit at 5 °C for 21 days storage (**A**) and transferred to 20 °C for 7 days (**B**). *N: fruit without packaging; 0: Fruit packed without perforation; 2: Fruit packed with perforation. Yellow arrows indicate chilling symptoms.

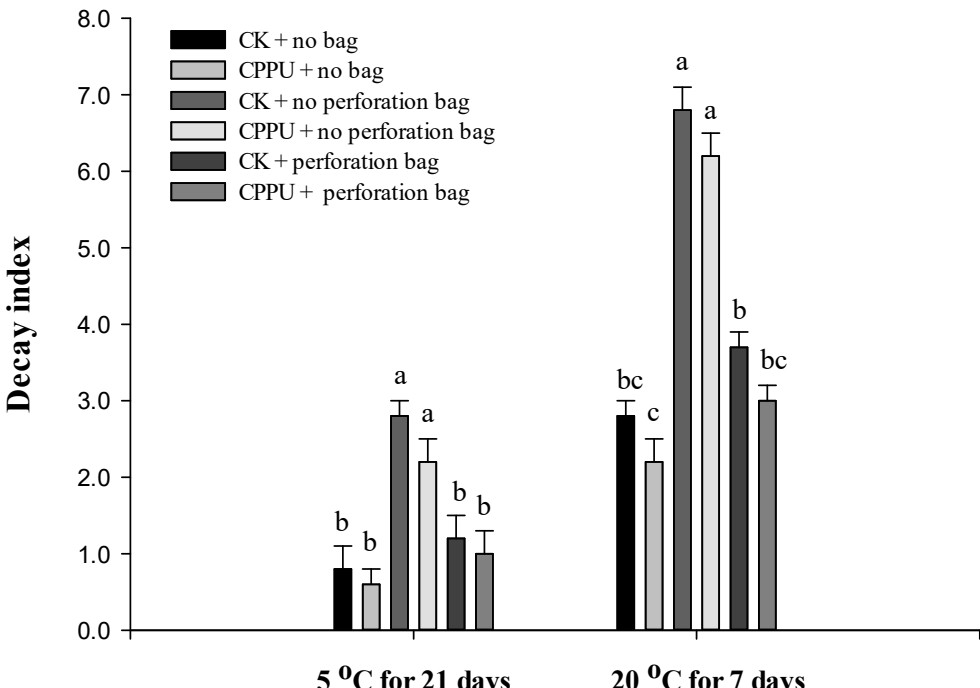

**Figure 4.** The effect of preharvest application of CPPU and perforation packaging on decay index (DI) of red-fleshed pitaya 'Da Hong' fruit at 5 °C for 21 days storage and transferred to 20 °C for 7 days. Bars represent means of ten replications ± SD (*n* = 4). Means with different lowercase letters are significant differences (*p* < 0.05).

### 3.5. Respiration Rate

The respiration rate was between 0.4 to 0.7 mL $CO_2 \cdot kg^{-1} \cdot h^{-1}$ in the fruit without packaging or when packed with perforated PE bags during storage at 5 °C, and was significantly higher in the fruit packed without perforated bags (between 1.6–1.8) (Figure 5). On the other hand, moving fruit to 20 °C storage resulted in a sharp increase in the respiration rate. The highest respiration rate was found in the CPPU treated fruit packed

without perforated bags (3.5 mL $CO_2 \cdot kg^{-1} \cdot h^{-1}$), followed by the control fruit without perforated packaging (2.8 mL $CO_2 \cdot kg^{-1} \cdot h^{-1}$). Although there was no significant difference between control and CPPU treated fruit, the respiration rate was found to be slightly higher in the CPPU treated fruit (Figure 5).

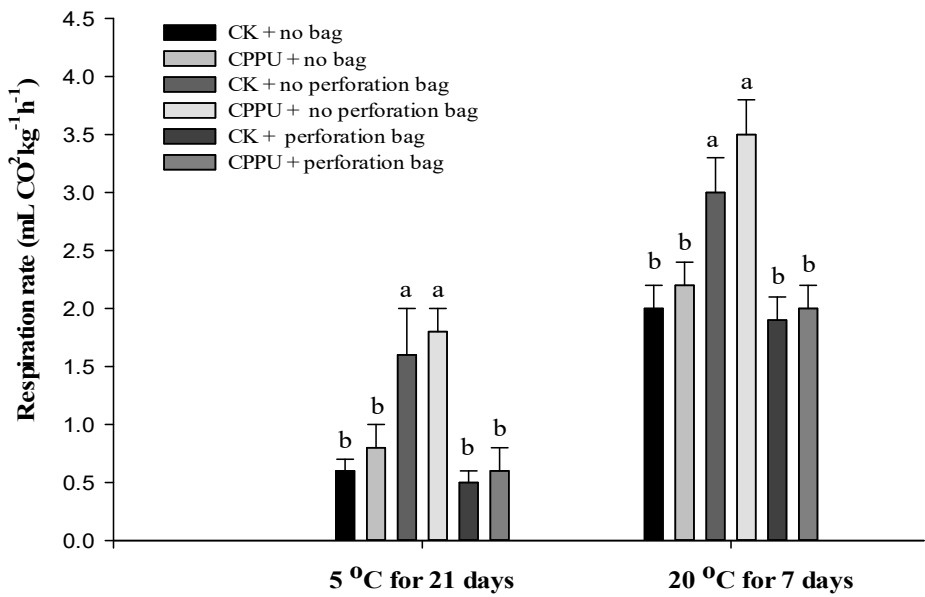

**Figure 5.** The effect of preharvest application of CPPU and perforation packaging on respiration rate of red-fleshed pitaya 'Da Hong' fruit at 5 °C for 21 days storage and transferred to 20 °C for 7 days. Bars represent means of ten replications $\pm$ SD ($n = 4$). Means with different lowercase letters are significant differences ($p < 0.05$).

## 4. Discussion

Although many reports have indicated that forchlorfenuron (CPPU) application increases fruit weight and fruit size [26,27], the usage of CPPU had no significant effect on fresh weight and fruit size in red-fleshed pitaya fruit at harvest in the current study (Table 1), which is similar to a previous report [12].

CPPU treated fruit not only had an increase in bract thickness and fruit firmness at harvest but also maintained this trend at 5 °C and at 20 °C storage period (Tables 1 and 2). Previous studies have shown that low temperatures could slow down the fruit softening of *H. polyrhizus* and *H. undatus* during storage [8–10,12,13]. Furthermore, Freitas and Mitcham [10] indicated that perforated packaging had no effect on fruit firmness, which is similar to our results that no PE bag packaging and packaging with/without perforated PE bags did not influence the bract thickness and firmness either in the CPPU treated or the control fruit (Table 2). Interestingly, it was suggested that microperforated films were effective in terms of firmness maintenance in strawberries [28] but the firmness of pomegranate arils decreased along with the number of perforations [5]. The possible explanations are that either the packaging material (e.g., low density polyethylene (LDPE)) or the number of perforations did not present the expected effects on pitaya fruit firmness and thickness in this study.

The weight loss of fresh fruit is mostly because of water loss, which increases during storage depending on storage conditions (e.g., temperature and packaging material). Perforated packaging has been proven to reduce weight loss significantly in yellow pitaya (*Selinicereus megalanthus* Shuman) [15] and in white-fleshed pitaya fruit *H. undatus* [10]. A similar finding was shown in this study when fruits were packed in the PE bag with/without perforation. This is due to the fact that fruit packed in PE bags may keep more relative humidity and experience less water loss than the fruit without packaging during storage (5 °C, 90 $\pm$ 5% relatively humidity). However, CPPU treated fruit without packaging had

more weight loss than the control fruit in storage (Figure 1), which supports a previous report [12]. This might be because CPPU is thought to mimic the role of cytokinins (CKs), which regulate stomatal activity and transpiration, leading to more water loss in the non-package condition during storage [29]. However, all fruits exhibited a rapid increase in weight loss during storage at 20 °C and 75 ± 5% relatively humidity because of the relatively high temperature, low humidity, and lack of packaging, which not only increased the respiration rate but also decreased the moisture content, leading to more dehydration [30].

In a previous study that white-fleshed pitaya fruit without perforated packaging stored at 5 °C for 20 days and then removed to 20 °C for 5 days had better appearance among all treatments [10]. It was reported that *H. undatus* had greener bracts when fruit were stored in the cold room at 10 °C than when stored at room temperature [31]. However, in the current study, regardless of whether or not PE bag packaging was used, the red-fleshed pitaya fruit treated with CPPU stored at 5 °C for 21 days exhibited greener bracts and a lower yellow index than the control fruit (Figure 2A,B). Although fruits were moved to 20 °C for 7 days, better appearance was observed in the CPPU treated fruit than in the control fruit (Figure 2A), with no influence of the perforated packaging. This could be explained by the fact that preharvest usage of CPPU caused more chlorophyll content in the bracts, resulting in greener color than the control fruit. Jiang et al. [12] once pointed out that preharvest application of CPPU resulted in significantly greener bracts in red-fleshed pitaya fruit. Another explanation is that CPPU acts as a blocker of cytokinin oxidase, which suppresses chlorophyll degradation [32,33].

Chilling injury occurs when fruits are in long-term storage at low temperatures. Previous studies reported that both *H. undatus* and *H. polyrhizus* pitaya fruits stored at 6 °C for two weeks and returned to room temperature (20 °C) had browning spots on the peel [8]. A minor chilling injury was observed in fruit stored at 5 °C for 21 days (Figure 3A) and severe CI symptoms were seen once fruits were moved to 20 °C for 7 days (Figure 3B) with/without PE bag packaging or packaging with/without perforation, which is in agreement with previous findings in *H. undatus* and *H. polyrhizus* sp., respectively [10,12].

A previous study showed that perforated packaging resulted in an increase in decay severity during different storage temperatures [10]. However, the highest decay incidence was observed in both CPPU treated and control fruit packed without perforated PE bags during postharvest storage (Figure 4). It is likely that PE bags without perforation created relatively high humidity, which favors the growth of microorganisms, although we did not count bacteria in this study. Another explanation might be that pathogens grow geometrically at 20 °C during storage period, which caused the severe decay index [31]. Furthermore, fresh fruit packed in the PE bag without perforations consequently produced higher $CO_2$ and lower $O_2$ levels during storage that most likely resulted in the severity of decay found in loquat fruit [16] and pomegranates [34].

Although the respiration rate had no statistical difference between the CPPU treated and the control fruit, the CPPU treated fruit had a slight increase in respiration rate compared with the control fruit (Figure 5). This respiration rate could be explained by the fact that more stomatal density was found in bracts of *H. undatus* [35] and the ripened pitaya fruit had few stomata on the peel [36,37]. In addition, the CPPU treated fruit had vivid green bracts in comparison to the control fruit, which possibly had more stomata in the current study. Since the function of CPPU could be related to regulating stomatal opening and transpiration [29], the CPPU treated fruit are likely to have more stomata, resulting in a higher respiration rate.

## 5. Conclusions

In conclusion, the preharvest application of CPPU and perforated packaging at 5 °C with 90% RH conditions resulted in better storage quality for 21 days. This combination resulted in significantly greener bracts, more firmness, and a lower decay ratio, and fewer chilling injuries, while excessive weight loss limited the fruit quality at 20 °C for 7 days. However, the effect of the combination of CPPU and perforated packaging may depend on

pitaya species, fruit maturity, bag material, and storage duration. To observe the useful effects of postharvest storage for different cultivated pitaya fruits, further studies are needed, especially on the interaction between different combinations and pitaya species.

**Funding:** This research was funded by the Council of Agriculture Executive Yuan, Taiwan, Grant No. (109AS-23.1.1-ST-a1), and the APC was funded by (109AS-23.1.1-ST-a1).

**Institutional Review Board Statement:** Not applicable.

**Informed Consent Statement:** Not applicable.

**Data Availability Statement:** Not applicable.

**Conflicts of Interest:** The author declares no conflict of interest.

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
