# Peer review of "Effect of Preharvest Application of CPPU and Perforated Packaging on the Postharvest Quality of Red-Fleshed Pitaya (Hylocereus polyrhizus sp.) Fruit"

_horticulturae, doi:10.3390/horticulturae7080253_

Round 1
Reviewer 1 Report
Dear Author,
manuscript is poorly written.
For review line numbering would be helpful. The manuscript has a lot of flaws, for example: mistake in afilliation of the Author, Abstract with empty place during the text. Blue color in keywords. Moreover, word keywords is written with a lowercase letter.
How the samples differed between "control without packaging" and "control withoud perforated packaging"? the second one were packed in unperforated package? Would be better to write all kinds of experiments in Table.
Reference 16 and 17 this is the same reference. Equations should be written like equations, not during the text (also cited and numbered for ex. eq. (1)) - page 3.
Fruit firmness - how about speed during penetrating probe?
All paragraphs should be numbered, for example 3. Results 3.1 Fruit physicochemical characteristics.
Table 1 and 2 should be insert in the main text after not before the paragraph of its first citation!
Values in tables and figures are means+/- SD, not SE?
Table 1 is cut.
Accuracy in tables should be the same. 0.1 or 0.01.
In Table 2 "N, 0, 2" are not defined.
In titles of Figures there is always "The Effect..." why is high letter inside?
"More weight loss from 1.0% to 3.5% sounds strange.
Figure 2 contains photos and curves - they sholud be two different Figures.
"Had a 1.1 yellow index compared to the control" - sounds strange
Figure 4 is too big. Bars have only +SD. Values -SD should be added.
Paragraph "Respiration rate" started in half of the sentence, which is the title of figure 5! Title of Figure 5 is cut.
In discussion is reference to [12] Table 1?
In the conclusions sentence "There will be several topics related how to prolong postharvest quality of pitaya fruit (...)" - is not a conclusion form this research.
Reference 1 should be adjusted to journal reference style.
Author Response
Please see the attachment, thank you!

Reviewer 2 Report
The manuscript entitled “Effect of pre-harvest application of CPPU and perforated packaging on the postharvest quality of red-fleshed pitaya (Hylocereus polyrhizus sp.) fruit is relevant for publication in Horticulturae after minor revision.
The author of the article undertook to investigate the effect of long-term refrigerated storage, pre-harvest application of CPPU and the method of packaging on maintaining the quality characteristics of red-fleshed pitaya fruits cv. 'Da-Hong. Obtained research results are interesting and properly described and discussed. The content of the article is valuable both in terms of science and practice, however, it requires minor revision.
Detailed comments:
Table 2 does not provide information on what N, 0 and 2 mean. Providing this information at this stage of the discussion, e.g. under Table 2 (similar to Figures 2 and 3) would contribute to a better understanding of the discussed results. The graphic side of the manuscript should be reorganized. e.g. the title of figure 5 is partly above and partly below the figure.
References
On page 11 of the article, the citation order is broken after reference 31. After citing 31 reference, citation 33 and 34 appear (page 12), while citation 32 is found on page 12.
On page 12, the last citation is quoted as 37, although in the References subsection the last entry is 38. This resulted from the double numbering of item 16 on page 13 (References). The citations should be organized in the text of the article and in References.
Author Response
Please see the attachment, thank you.

Round 2
Reviewer 1 Report
Dear Author,
thank you for improving and correcting the manuscript, in present form article sounds better.
Author Response
Please see the attachment, thank you.
